# Self-encapsulated ionic fibers based on stress-induced adaptive phase transition for non-contact depth-of-field camouflage sensing

Ying Liu[1,2,7], Chan Wang[1,2,7], Zhuo Liu[1,3,7], Xuecheng Qu[1,2], Yansong Gai[1], Jiangtao Xue[1,4], Shengyu Chao[1,2], Jing Huang[1,2], Yuxiang Wu[1,5], Yusheng Li[1,6], Dan Luo ®[1,2] ✉ & Zhou Li ®[1,2] ✉

Ionically conductive fibers have promising applications; however, complex processing techniques and poor stability limit their practicality. To overcome these challenges, we proposed a stress-induced adaptive phase transition strategy to conveniently fabricate self-encapsulated hydrogel-based ionically conductive fibers (se-HICFs). se-HICFs can be produced simply by directly stretching ionic hydrogels with ultra-stretchable networks (us-IHs) or by dip-drawing from molten us-IHs. During this process, stress facilitated the directional migration and evaporation of water molecules in us-IHs, causing a phase transition in the surface layer of ionic fibers to achieve self-encapsulation. The resulting sheath-core structure of se-HICFs enhanced mechanical strength and stability while endowing se-HICFs with powerful non-contact electrostatic induction capabilities. Mimicking nature, se-HICFs were woven into spider web structures and camouflaged in wild environments to achieve high spatio-temporal resolution 3D depth-of-field sensing for different moving media. This work opens up a convenient route to fabricate stable functionalized ionic fibers.

The emergence of soft materials has greatly boosted the development of flexible electronics to gather physiological and environmental information[1,2]. Fibers, as one of the most important participants, demonstrate unique advantages with large specific surface area, high adaptability and good customizability with other functional units[3,4], with a wide range of promising applications in transistors, fiber-based circuitry, sensing networks, and energy harvesting or storage devices[2,5]. Among them, ionic conductive hydrogel fibers have emerged as an important candidate on account of its unique advantages including transparency, large aspect ratio, great weavability, and excellent mechanical flexibility. In general, the manufacture of hydrogel-based fibers mainly based on spinning techniques, including wet/dry/gel spinning, electrospinning[6], direct ink writing[7], draw spinning[8], and microfluidics[9], but the following challenges still remain: (i) The hydrogels and their precursor solutions are poorly spinnable, making it difficult to fabricate meters-sized or longer fibers on a large

¹Beijing Institute of Nanoenergy and Nanosystems, Chinese Academy of Sciences, Beijing 101400, China. ²School of Nanoscience and Engineering, University of Chinese Academy of Sciences, Beijing 100049, China. ³Key Laboratory of Biomechanics and Mechanobiology, Ministry of Education, Beijing Advanced Innovation Center for Biomedical Engineering, School of Engineering Medicine, Beihang University, Beijing 100191, China. ⁴School of Life Science, Institute of Engineering Medicine, Beijing Institute of Technology, Beijing 100081, China. ⁵Department of Health and Kinesiology, School of Physical Education, Jianghan University, Wuhan 430056, China. ⁶National Clinical Research Center for Geriatric Disorders, Xiangya Hospital, Central South University, Changsha 410008, China. ⁷These authors contributed equally: Ying Liu, Chan Wang, Zhuo Liu. ✉e-mail: luodan@binn.cas.cn; zli@binn.cas.cn

scale. (ii) The conventional strategies are energy-intensive, requiring many cumbersome production procedures and large solvent consumption, which is not in line with the trend of green chemistry[10,11].

In addition to the difficulties of large-scale manufacturing, exposure of ionic conductive hydrogel fibers to the environment can lead to severe dehydration and deionization due to the large specific surface area, and their poor long-term stability further limits their widespread application. Currently, the introduction of high hygroscopic salts to reduce vapor pressure by forming hydrated ions with water molecules has been proved to be an effective approach to improve the dehydration resistance of hydrogel fibers[1]. However, it will swell and deswell in response to ambient humidity, leading to unstable electrical properties[12]. Another approach is to encapsulate hydrogel fibers by wrapping, covering, and coating other hydrophobic materials to reduce the water evaporation and prevent interconnection of bare conductive hydrogel fibers[8,13]. However, most encapsulation strategies face issues such as modulus mismatch and weak interfacial adhesion between the core and encapsulation layers, leading to peeling and abnormal fracture during the service period[13]. This dilemma emphasizes the importance of developing an environmentally friendly and efficient strategy to fabricate ionic conductive hydrogel fibers with stable encapsulated structures.

Herein, we developed a stress-induced adaptive phase transition strategy for the sustainable fabrication of highly stable functional self-encapsulated hydrogel-based ionically conductive fibers (se-HICFs), consisting of only two steps: stress-induced stretching/dip-drawing molding, and self-encapsulation triggered by adaptive phase transition. As the precursor of se-HICFs, ionic hydrogels with ultra-stretchable polymer network were preferentially synthesized based on the principles of long/short polymer complementarity and dynamic physical interactions. The acquired ultra-stretchable ionic hydrogels (us-IHs) possessed an ultimate tensile deformation of >1,900,000 times, and the microfibers could be formed either by directly stretching from us-IHs or dip-drawing from the molten us-IHs. The adaptive phase transition occurred concomitantly, in which water molecules migrated outward from the surface of the hydrogel fibers under stress induction, eventually forming an anhydrous encapsulation structure. The self-encapsulated layer formed by the phase transition greatly enhanced the mechanical strength of se-HICFs by >1200 times, and effectively prevented the loss of internal moisture. In addition, the ultrafine self-encapsulated sheath-core structure also endowed se-HICFs with strong electrostatic induction capability for efficient non-contact 3D depth-of-field sensing. se-HICFs can be fabricated on a large scale and flexibly woven into a biomimetic spiderweb structure for monitoring the movements of insects, unmanned aerial vehicle (UAV), and humans in natural environments, showing great potential in smart agriculture and camouflage monitoring for homeland security.

## Results

### The principle of fabricating se-HICFs by using us-IHs as precursors

Ultra-stretchable ionic hydrogels, as precursors, are the basis for the successful preparation of se-HICFs, and the process of fabricating se-HICFs based on us-IHs includes the following two steps (Fig. 1a). Firstly, ionic hydrogel fibers with a diameter of micron-scale were obtained by multistage stretching of us-IHs or direct dip-drawing from molten us-IHs. The fibers molded in the first step were highly stretchable and

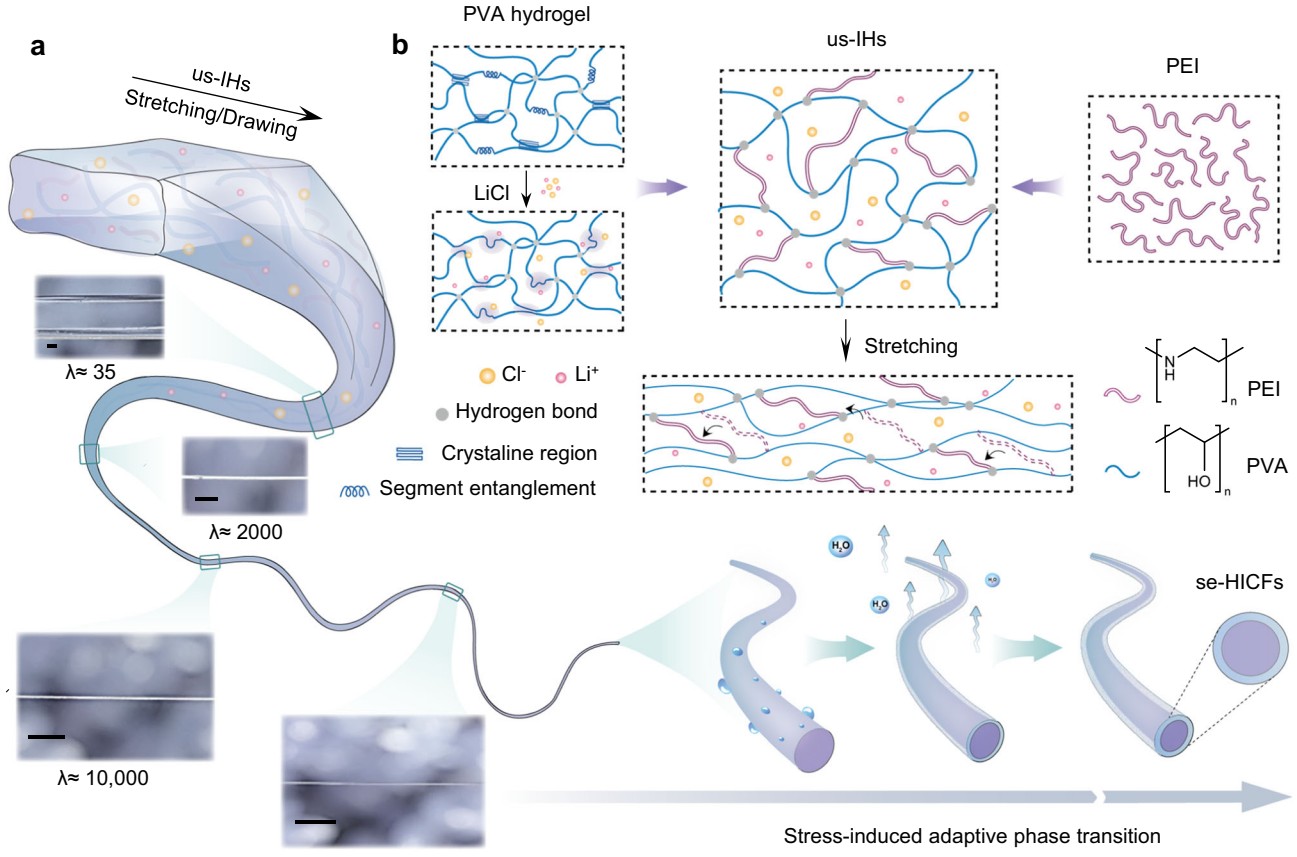

**Fig. 1 | Design principle of the ultra-stretchable ionic hydrogel (us-IHs) and self-encapsulated hydrogel-based ionically conductive fibers (se-HICFs).**
**a** Schematic illustration about the design principle and the ultra-stretchability of us-

IHs. Scale bar: 2.5 mm. **b** Schematic illustration about the fabrication process of the se-HICFs based on stress-induced adaptive phase transition strategy. Insets show are photographic images of us-IHs in different stretching ratio.

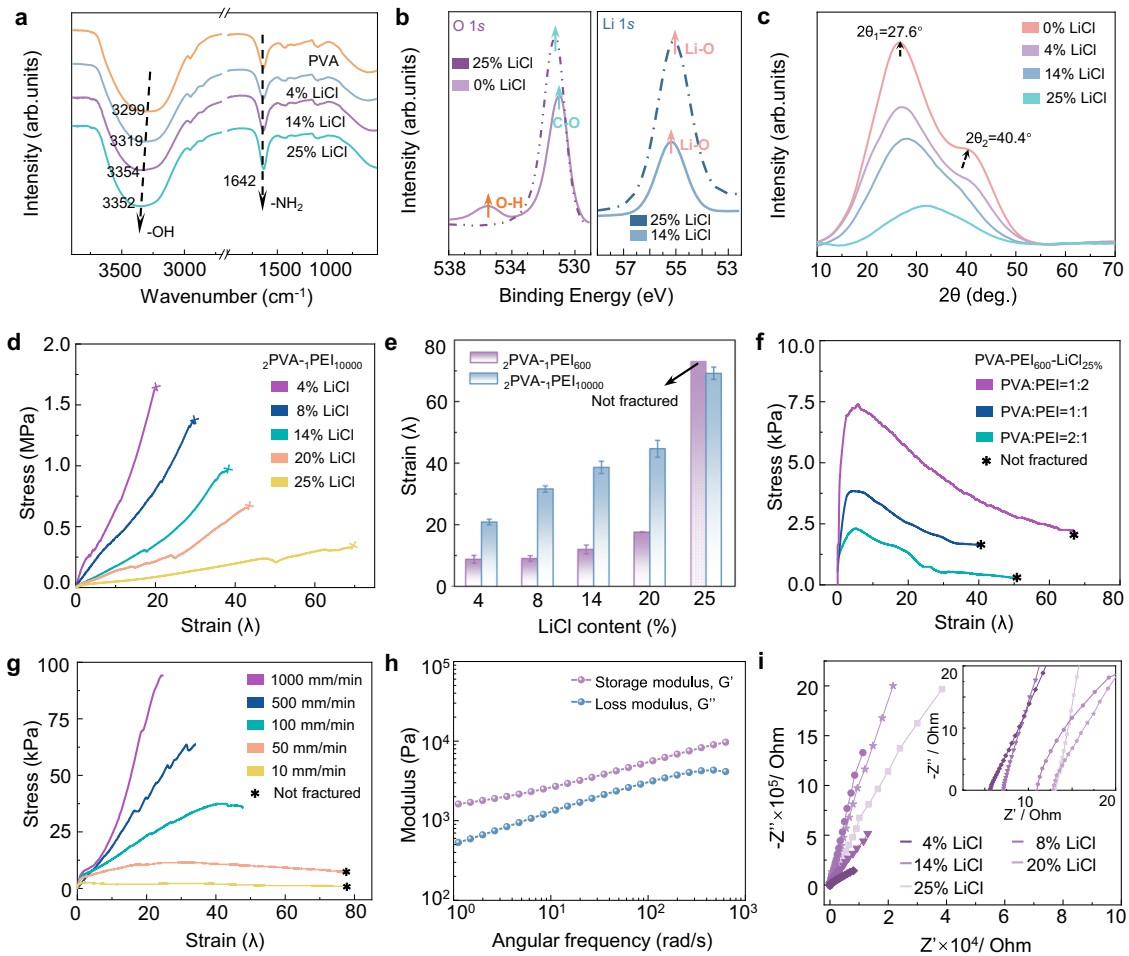

**Fig. 2 | Characterizations, mechanical and electrical properties of us-IHs. a** FTIR spectrums of pure PVA and LiCl doped hydrogels. **b** XPS spectra on O 1$s$ and Li 1$s$ of hydrogels. **c** X-ray diffractogram of PVA-PEI hydrogel and LiCl doped hydrogels. **d** Stress-strain curves of the $_2$PVA-$_1$PEI$_{10000}$ with different LiCl content. **e** Tensile strain of $_2$PVA-$_1$PEI$_{10000}$ and $_2$PVA-$_1$PEI$_{600}$ with different LiCl content. **f** Stress-strain cures of $_2$PVA-$_1$PEI$_{600}$, $_1$PVA-$_1$PEI$_{600}$ and $_1$PVA-$_2$PEI$_{600}$. For all samples in Fig. 2d–f,

were shaped in size of $30 \times 10 \times 1$ mm$^3$ and tested with a loading rate of 50 mm·min$^{-1}$. **g** Stress-strain curves of us-IHs with different loading rates. **h** Frequency-dependent oscillatory rheology of us-IHs ($\tau = 1\%$, 25 °C). **i** The electrochemical impedance spectra (EIS) of the LiCl doped hydrogels. The error bar for each data point in (**e**) is standard deviation based on 3 parallel measurements.

flexible, but had weak mechanical strength (Supplementary Movie 1). Concomitantly, the stress-induced directional migration and evaporation of water molecules on the surface of the ionic hydrogel fiber facilitated the transformation of surface layer to form a tight encapsulation layer. The stress-induced phase transition that occurs on the fiber surface was self-adaptive, and the self-encapsulation process continued until the equilibrium state was reached, then the thickness of the self-encapsulation layer remained constant, indicating the formation of se-HICFs with a sheath-core structure. The mechanical strength of acquired se-HICFs reached the order of MPa-level, which was more than one thousand times higher than that of ionic hydrogel fibers before self-encapsulation. In addition, the self-encapsulated layer caused by the phase transition also prevented the internal moisture loss, ensuring the long-term stability of se-HICFs.

## Design, optimization, and chemical/mechanical/electrical characterizations of us-IHs

The key to design us-IHs is the construction of dynamic cross-linked networks with large energy dissipation abilities[14]. In our system, the us-IHs consisted of polyvinyl alcohol (PVA), polyethyleneimine (PEI) and LiCl, where long/short polymer chains formed a dynamic cross-linked network via hydrogen bonds (Fig. 1b). The long PVA chains with high mechanical strength behaved as the backbone of the polymer network,

while the short-chain PEI enhanced chain mobility and reduced internal frictions, leading to greater stretchability[15]. The ion-dipole interactions between Li$^+$ and hydroxyl functional groups (-OH) in PVA polymer chains could disrupt the inter/intramolecular hydrogen bonds between the crystalline regions of PVA, thereby improving the segment mobility[16,17]. The Fourier Transform Infrared Spectroscopy (FTIR) showed that with the increase of Li$^+$ content, the stretching vibration absorption peak of -OH in PVA chains moved from 3299 to 3352 cm$^{-1}$, indicating that Li$^+$ interacted with -OH in the network (Fig. 2a)[16]. Additionally, X-ray photoelectron spectroscopy (XPS) further demonstrated the interactions between Li$^+$ and O in -OH due to the disappearance of peak at BE = 535.48 eV assigned to O-H in O 1$s$ spectrum (shift to lower energy and overlap with the peak assigned to C-O) with increasing LiCl content, and the peaks in Li 1$s$ spectrums were assigned to Li-O indicating the interactions from ion-dipole interactions (Fig. 2b)[18]. According to the X-Ray Diffraction (XRD), the area of PVA crystalline regions gradually decreased and eventually turned into a uniform amorphous phase, which indicated that more PVA polymer chains were released from the tightly packed crystalline regions (Fig. 2c)[19,20]. When external forces are applied, us-IHs dissipated energy through segment slippage and dynamic equilibrium of hydrogen bond breakage/reorganization. As a result, it could withstand extreme puncture without rupture (Supplementary Movie 2).

Compared with the reported polymeric materials with high stretchability based on typical strategies, such as nanocomposite structures, double networks and dynamic crosslinking systems (including hydrophobic associations, metal-ligand interactions, host-guest interactions, ionic bonds, and hydrogen bonds networks), us-IHs presented the highest stretching limits, broadening the range of stretchable materials (Supplementary Fig. 1 and Supplementary Table 1).

The stretchability and conductivity of us-IHs could be optimized by adjusting the amount of LiCl added, the molecular weight of PEI and the mass ratio of PVA/PEI. The tensile strain-stress curves of us-IHs showed that the introduction of LiCl led to greater stretchability and weaker mechanical strength. When the molecular weight of PEI and the mass ratio of PVA/PEI were fixed, the tensile strain ($\lambda = \Delta L/L_0$) rose from 20 to 69 as the LiCl content increased from 4 % to 25 %, while the tensile strength dropped by 19.5% of the original value at the same time (Fig. 2d). The polymer chain length of PEI was also one of the major factors affecting the mechanical properties of ionic hydrogels, since shorter PEI chains provided better mobility[21]. We investigated the tensile behavior of ionic hydrogels with PEI molecular weight of 10,000 and 600 at different LiCl contents[22]. It should be noted that the deformation of the sample with molecular weight of 600 suddenly increased and exceeded that of the sample with molecular weight of 10000 when LiCl was added at 25%, indicating that when the adequate amount of LiCl was incorporated into the polymer network, intermolecular/intramolecular hydrogen bonds leads to sufficient disentanglements of the PVA chains, so that more low molecular weight PEI can participate in the crosslinking network and promote the stretchability (Fig. 2e, Supplementary Fig. 2 and Supplementary Table 2)[23]. Subsequently, the effect of PVA/PEI ratio on the mechanical properties of ionic hydrogels was further explored. Regardless of the PVA:PEI ratio of 2:1, 1:1 or 1:2, the as-prepared hydrogels exhibited ultra-stretchability and did not fracture within the test range of Mark-10 system (Fig. 2f and Supplementary Fig. 3). The mass ratio of PVA/PEI mainly affected the mechanical strength of the ionic hydrogels, because more PVA chains provided stronger network backbone. Overall, the $_2PVA$-$_1PEI_{600}$-$LiCl_{25}$ (us-IHs) exhibits the highest level of stretchability, making it the optimal ratio for the subsequent studies.

The tensile deformation of $_2PVA$-$_1PEI_{600}$-$LiCl_{25}$ exhibited a dependence on the stretching speed. As shown in Fig. 2g, us-IHs did not fracture when the loading speed was within 50 mm/min. Once the loading speed exceeded this range, us-IHs rapidly underwent elastic fracture quickly because the migration rate of the polymer segments cannot keep up with the stretching rate[24]. Furthermore, the frequency-dependent oscillatory rheology of us-IHs demonstrated that the storage modulus (G′) was superior to the loss modulus (G″) in the frequency range from 1 to 1k Hz (Fig. 2h), indicating the elastic behavior of us-IHs and the formation of the cross-linked network[25]. The conductivity of us-IHs mainly derived from Li⁺, Cl⁻ and electrolytic anions of PEI. The electrochemical impedance spectra (EIS) of us-IHs showed that its conductivity gradually increased from 1.5 to 3.6 S/m when the content of LiCl rose from 4% to 25% (Fig. 2i and Supplementary Fig. 4)[26]. Moreover, us-IHs with lower molecular weight PEI had the better conductivity, mainly due to the looser cross-linked network that facilitated the free migration of ions (Supplementary Fig. 5)[19,27].

## Stress-induced stretching/dip-drawing molding of us-IHs into ionic fibers

The ionic hydrogel fibers can be fabricated by direct stretching of bulk us-IHs at room temperature. In order to explore the effect of stress on the morphology of fibers, us-IHs with the most excellent mechanical and electrical conductivity properties were selected for multistage stretching test (Fig. 3a)[28]. The total stretchability ($\lambda$) of the us-IHs could be calculated by the production of the stretching ratios at each stage ($\lambda_1, \lambda_2, \lambda_3, \lambda_4$), that is, $\lambda = \lambda_1 \times \lambda_2 \times \lambda_3 \times \lambda_4 = 71 \times 36 \times 42 \times 18 = 1,932,336$[29]. Correspondingly, bulk us-IHs could be stretched into micro-scale

fibers with diameters of 846, 380, 88 and 20 μm at different stretching stages (Fig. 3b and Supplementary Movie. 3).

Interestingly, besides stretching molding, ionic hydrogel fibers can also be directly dip-drawing from the molten us-IHs, and the fiber diameter could be controlled by the extraction process (Fig. 3c and Supplementary Movie 4). Dip-drawing spinning is efficient and does not require complicated spinning equipment[30,31]. The physically cross-linked networks and short PEI chains of us-IHs gave rise to a temperature-induced increase in segment mobility and disentanglement, which made it more accessible to draw the fibers directly from molten us-IHs[30]. Furthermore, the long PVA chains in the network facilitated the chain realignment and were less prone to fracture during the fiber spinning. The us-IHs in the molten state exhibited a temperature-dependent transition in viscoelastic behavior, as indicated by the G′/G″ in rheological test (Fig. 3d). With the increase of temperature, us-IHs switched from elastic-dominated behavior to viscous-dominated behavior[32]. Figure 3e showed the loss factor (tan δ = G″/G′), which is fundamentally related to the chain entanglement effect and is also critical for the spinning capability[33]. Compared with us-IHs at room temperature, the approximate values of tan δ of us-IHs at 50, 65 and 80 °C were mostly in the range of 0.6 to 1.1, suggesting that molten us-IHs was highly spinnable[33]. As shown in Fig. 3f, the needles fixed on an acrylic rod were immersed into the us-IHs melt, and then lifted at a certain speed to spin out the fiber through the adhesion force between the tip and the melt. Supplementary Movie 5 showed the entire process of direct spinning ionic hydrogel fibers by dip-drawing from molten us-IHs.

The correspondence between different process parameters and spinning controllability was further discussed. The needle tips size played a vital role in controlling the initial contact area between molten us-IHs and needles, which affected the diameter of the spun fiber. As shown in Fig. 3g, the diameter of the main fibers increased from 200 to 600 μm using needles with different size (400, 600 and 1600 μm) for dip-drawing. Furthermore, the melt temperature also affected the fiber size for spinning (Fig. 3h). As the melt temperature increased by 60 °C, 70 °C and 80 °C, the main fiber size tended to decrease which may be attributed to the difference in segmental mobility at different temperatures, making chain realignment occur more easily. The extraction speed was another important parameter controlling the fiber size (Fig. 3i). The fiber size decreased slightly with increasing extraction speed (500, 750 and 1000 mm/min), indicating the higher extraction speed could intensify the melt necking behavior and polymer chain alignment[33]. Fibers with a length of several meters could be immediately molded at room temperature after dip-drawing from the us-IH melt and then wrapped on a reel, showing the potential of large-scale preparation (Fig. 3j). The fibers prepared by dip-drawing could also be secondarily stretched, and the diameter of the fibers could be as small as few tens of microns (Fig. 3k and Supplementary Movie 6). Surprisingly, the hydrogel fibers gradually showed the beads-on-a-string structure after stretching, similar to natural spider silk (Fig. 3l).

## The formation of self-encapsulation layers caused by adaptive phase transitions

During the stress-induced conversion of us-IHs into ionic hydrogel fibers with diameter of micrometers, the adaptive phase transition occurred on the surface of the fibers. The shrinkable polymer network after stretching induced the outward directional migration of water molecules. Due to the large specific surface area of ionic hydrogel fibers, the water molecules migrated to the fiber surface were then evaporated rapidly, eventually leading to a phase transition of the surface layer to form a self- encapsulated structure. Specifically, the self-encapsulation induced by the adaptive phase transition involved three processes: (i) directional migration of water molecules to the outside of the fiber, (ii) the evaporation of water molecules on the fiber surface, and (iii) reaching the equilibrium state of water evaporation

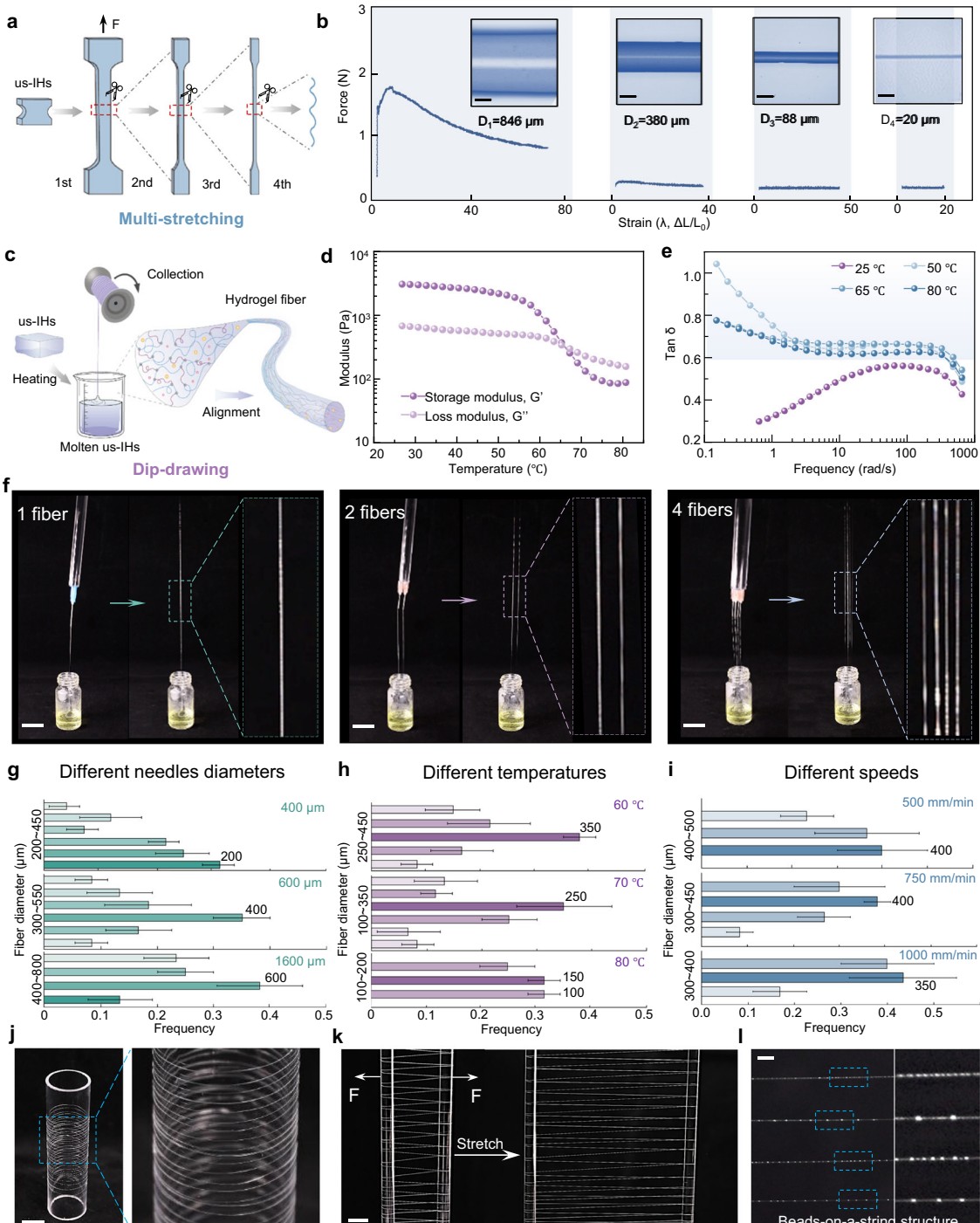

**Fig. 3 | Stress-induced stretching/dip-drawing of us-IHs into ionic fibers.**
**a** Schematic illustration of multistage stretching. **b** Force-strain curves of us-IHs with the loading rate of 30 mm·min⁻¹. The force-strain tests contain four stages. In the second stage, the sample was cut from the stretched sample in first stage. The stretching tests in third and fourth stage were performed in same way. Insets showed the samples after tensile stretching in every stage. Scale bar: 100 μm. **c** Schematic illustration of the dip-draw spinning of the ionic hydrogel fibers. **d** Temperature-dependent oscillatory rheology of us-IHs ($\tau = 1\%$, $\omega = 10$ rad/s). **e** The loss factor (tan δ) as a function of frequency for hydrogel melt with varying temperatures. **f** Photographs during dip-drawing spinning of the ionic hydrogel fibers with different needles. Scale bar: 2 cm. **g** Effects of the needle's diameters (**h**) molten us-IHs temperatures and (**i**) drawing speed. **j** Photograph of ionic hydrogel fibers after dip-draw spinning. Scale bar: 2 cm. **k** Photograph shows the great stretchability of ionic hydrogel fibers. Scale bar: 1 cm. **l** Photographs of beads-on-a-string structure in ionic hydrogel fibers. Scale bar: 2 mm. The error bar for each data point in (**g**, **h**, **i**) is standard deviation calculated based on 20 parallel measurements.

and absorption. As shown in Fig. 4a, the outward directional migration of water molecules in the first stage was driven by network shrinkage under tensile stress and the moisture content gradient between the hydrogel and the environment[34,35]. The outward-migrating water molecules gradually converged on the surface of the ionic hydrogel

fiber and fused to form the water drop over a period of 4 h (Fig. 4b)[36]. This phenomenon occurred mainly because the outward migration speed of water was much faster than the evaporation rate of water, and the excess water condensed into beads under the effect of the surface tension, which could also explain the spider silk-like beads-on-a-string

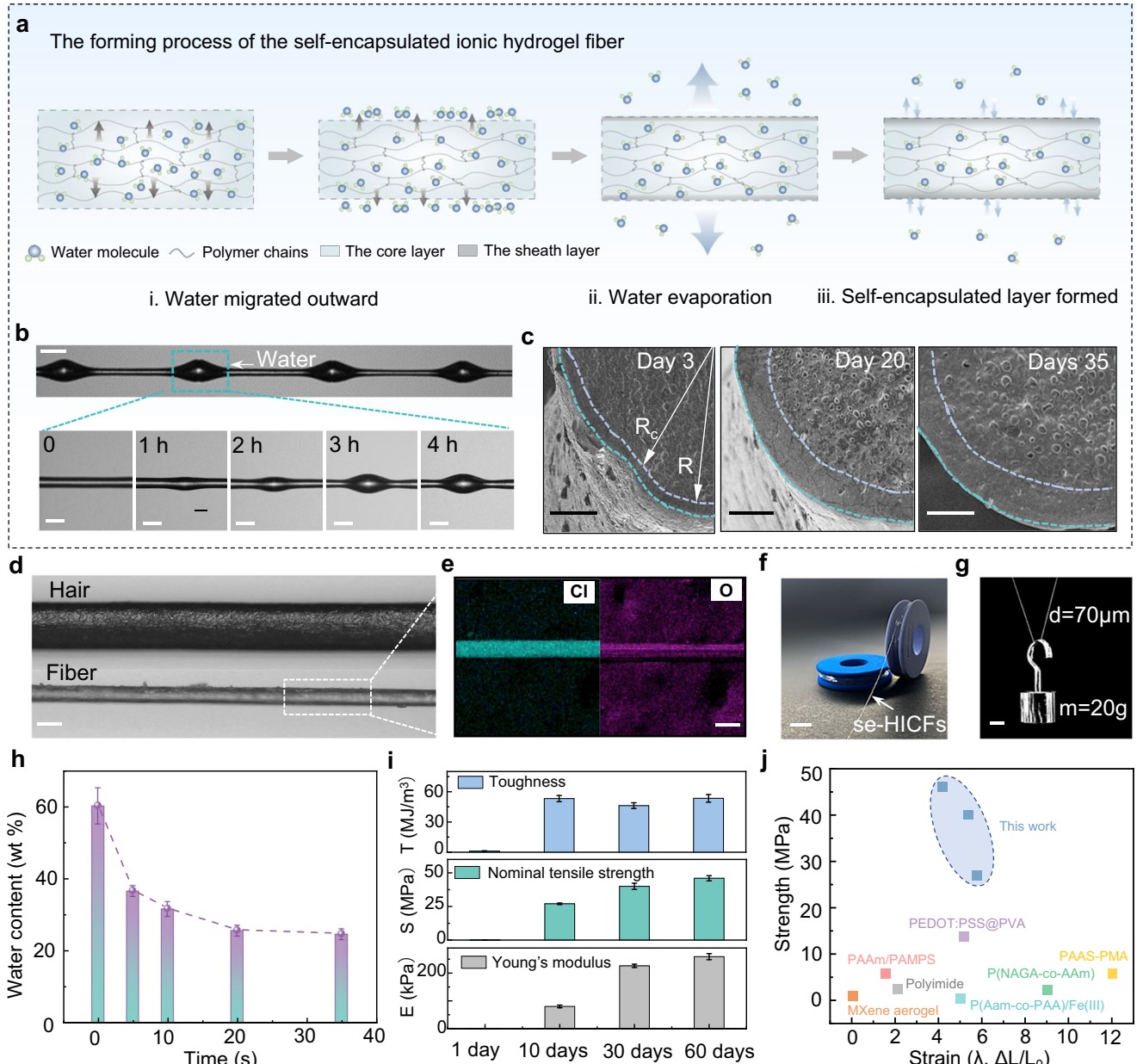

**Fig. 4 | The stress-induced adaptive phase transition process and mechanical properties of se-HICFs. a** Mechanism of stress-induced adaptive phase transition for the preparation of se-HICFs. **b** Optical images about the process (within 4 h) of water seeping from the fiber. Scale bar: 100 μm. **c** SEM images of the se-HICF's cross-section, showing sheath-core structures. Scale bar: 25 μm. **d** Optical images showed the se-HICF is thinner than the human hair. Scale bar: 50 μm. **e** The Cl and O elemental mapping images of se-HICF. Scale bar: 50 μm. **f** se-HICF was wound on the thread spool. Scale bar: 1 cm. **g** se-HICFs with a diameter of 70 μm could lift a weight of 20 g. Scale bar: 0.5 cm. **h** The variation of se-HICF's water content within 35 days. **i** Toughness, Nominal tensile strength, and Young's modulus of se-HICFs within 60 days. **j** A comparison between se-HICFs and previously reported functional fibers in terms of tensile strength and strain. The details are summarized in Supplementary Table 2. The error bar for each data point in (**h**, **i**) is standard deviation calculated based on 3 parallel measurements.

structure in Fig. 3l. Subsequently, the outward migration of water decreased and the process entered the second stage dominated by water evaporation. During this stage, water evaporated from the surface of ionic hydrogel fibers at room temperature, gradually underwent a phase transition and formed a sheath layer to prevent further evaporation of internal water molecules. The scanning electron microscopy (SEM) images further exhibited the formation process of self-encapsulation (Fig. 4c). The freeze-dried se-HICFs exhibited a typical sheath-core structure, consisting of a sheath layer with dense morphology due to dehydration and a water-rich porous core layer. It is worth mentioning that the phase transition leading to the self-encapsulated structure was adaptive. In the early stage, the sheath

layer formed by the phase transition of the surface ionic hydrogel fiber was not sufficient to suppress the volatilization of internal water, and the phase transition further extended into the interior until reaching the equilibrium state of water evaporation and water absorption[37]. As the surface water evaporated, the sheath layer formed by the phase transition became thicker, and the ratio of core radius to fiber radius (Rc/R) was $0.90 \pm 0.01$ $0.83 \pm 0.04$ and $0.83 \pm 0.02$ on Day 3, Day 20 and Day 35, respectively. Notably, from Day 20 to Day 35, the thickness of sheath layer remained unchanged for 15 days, indicating that the adaptive phase transition was complete. These findings were further confirmed by the thermo-gravimetric analysis (TGA), which showed that the water content of se-HICFs remained essentially the same from

Day 20 to Day 35 (Fig. 4h and Supplementary Fig. 6), demonstrating that se-HICFs reached an equilibrium state of phase transition. This is mainly attributed to the strong hydration of high-content lithium ions in se-HICFs and the formed self-encapsulating shell that together inhibited further evaporation of water[38-40], ensuring the long-term stability of the sheath-core structure (Supplementary Fig. 7)[27].

The morphology, strength and stability of se-HICFs were further explored in depth. The average diameter of se-HICFs was in tens of micrometers, thinner than a human hair (Fig. 4d). Energy Dispersive X-Ray Spectroscopy (EDX) showed that the Cl and O elements were uniformly distributed on the surface of se-HICFs without obvious defects (Fig. 4e). Furthermore, se-HICFs with a diameter of 70 μm could lift an object of 20 g, >60,000 times its own weight (Fig. 4f, g). The dense self-encapsulated sheath layer endowed the fibers with stronger mechanical properties (Fig. 4i and Supplementary Fig. 8). Toughness, nominal tensile strength and Young's modulus behaved significant enhancement in the first 10 days and continued their increasing trend within 60 days. The se-HICFs in the 60th day could withstand >400% tensile deformation, and the toughness, nominal tensile strength and Young's modulus was 53 MJ/m$^2$, 46 MPa, and 259 kPa, respectively. In addition, se-HICFs could still recover to its original length after 1000 tensile loading-unloading cycles at 200% strain, indicating its great recovery and fatigue resistance (Supplementary Fig. 9). Compared to other similar reports, the se-HICFs standout for its superior comprehensive properties (Fig. 4j and Supplementary Table. 3)[8,11,25,41-44]. Additionally, we also assessed the stability of the se-HICFs in different humidity environments[45]. The increase in relative humidity resulted in enhanced ductility, reduced tensile strength, and higher elongation of se-HICFs (Supplementary Fig. 10); However, even at high humidity (RH 85%), se-HICFs still had great mechanical property with the tensile strength of 16.8 MPa and a large tensile deformation of 843%. Even so, se-HICFs did not undergo swelling or structural collapse after being immersed in water for 60 min, and maintained great mechanical properties, highlighting the robustness and stability of the self-encapsulated layer (Supplementary Fig. 11).

## The non-contact 3D depth-of-field sensing ability of se-HICFs

Considering long-term reliability, mechanical robustness and large specific surface area, se-HICFs are ideal stretchable self-powered sensors for non-contact 3D depth-of-field sensing based on electrostatic induction. In the sheath-core structure of se-HICFs, the self-encapsulated sheath was used as an insulating layer (Supplementary Fig. 12), and the conductive core based on ionic hydrogel served as an electrode, forming a structure similar to a single-electrode triboelectric nanogenerator (Fig. 5a)[46,47]. The detailed working mechanism of the se-HICFs was schematically demonstrated in Fig. 5b and Supplementary Fig. 13, 14[48]. When a moving object with positive charges approached the se-HICF, Cl$^-$ ions were absorbed to the side of core layer close to the moving object, while Li$^+$ ions were repelled to the other side; as the moving object approached, free electrons flowed in to balance the accumulated cations on the opposite side and generated an output electrical signal as they passed through the external circuit. Conversely, when a positively charged object moved away from the se-HICF, the ions gradually returned to the free distribution state and generated a reverse output signal in the circuit. This is the complete cycle for se-HICF-based sensing. The corresponding COMSOL simulation schematic diagram further confirmed the above non-contacting sensing principle through the calculated potential distribution (Fig. 5c)[49].

The se-HICFs possessed strong electrostatic induction capability. As shown in Fig. 5d, the open-circuit voltages ($V_{oc}$) slightly increased as the diameter of the se-HICFs decreased, suggesting that larger resistance was more beneficial to the noncontact sensing performance due to the stronger shielding properties. The non-contact 3D depth-of-field

sensing performance of a single se-HICFs was systematically investigated by quantitatively analyzing the sensing signals generated from different distances and distinct reference object areas (Fig. 5e, f). When the distance decreased from 250 mm to 5 mm, $V_{oc}$ increased from 0.024 V to 2.3 V. Even when the area of sensing object was only 1 × 1 cm$^2$, the $V_{oc}$ could still reach 0.5 V. Furthermore, the moving speed of objects and the difference in materials also influenced the electrostatic induction process of the se-HICFs. Figure 5g showed that the $V_{oc}$ frequency generated by se-HICFs increased from 0.5 Hz to 2.5 Hz as the object moved faster, indicating that the speed of motion was proportional to the polarization potential. The se-HICFs also could identify different approaching materials with difference surface charges (Fig. 5h). Stability and reliability are important characteristics of sensors[50]. se-HICFs maintain high non-contact sensing sensitivity after 60 days of use (Fig. 5i), Furthermore, se-HICFs also exhibited great electrostatic induction properties even in high-humidity environments (RH 85%), revealing their stability and broad application prospects as non-contact self-powered sensors (Supplementary Fig. 15).

## Programmatic weaving of se-HICFs to form an ISW for camouflaged depth-of-field monitoring

The se-HICFs could be programmatically woven into 2D or 3D biomimetic structures to enhance sensing capabilities and meet different environmental requirements. In nature, the spiders spin adhesive, stretchable and translucent trapping threads on structurally strong frame threads that retain their mechanical strength even in inclement weather and provide the spiders with agile maneuverability[51]. Inspired by the natural spider web structure, transparent, ultra-fine, stretchable se-HICFs with spider silk morphology could be fabricated into spider web and camouflaged in natural environments (Fig. 6a). The mechanically robust se-HICFs were first cross-fixed into the radial threads to form the frame, and then the remaining se-HICFs were glued on the frame as spiral threads. The se-HICFs with a length of ~8 m prepared under laboratory conditions could be weaved into a giant ionic conductive spider web (ISW) with a diameter of ~1.75 m, showing the potential for large-scale fabrication (Fig. 6b). ISW showed great camouflaged ability in different environments, such as grass, fallen leaves, gravel and rubber track (Fig. 6c). Furthermore, the ISW could withstand the high-speed impact formed by a solid ball, indicating its excellent mechanical robustness (Fig. 6d). Transcending the spider web, the ISW could detect the motion behavior of different insects (butterfly, beetle and ladybug) approaching and touching the spider web-like structure through 3D depth-of-field sensing (Fig. 6e and Supplementary Fig. 16). As the insects flew around the ISW, their real-time motion signals could be monitored (Fig. 6f). The induced potential changes when insects alighting/flying away from the ISW surface were closely related to the size of the insects. The highest voltage was produced by the movement of the butterfly, followed by beetle and ladybug (Fig. 6g). Additionally, differences in the waveforms of the induced electrical signals also indicated the distinct locomotion patterns of insects. Taking the movement of a butterfly as an example, compared with flying around the ISW, landing on the ISW and flapping wings produced a more random, low-intensity, and high-frequency signal with a higher baseline potential (Fig. 6h). Besides, when the butterfly flew around the ISW, its flight altitude can also be monitored by the magnitude of the induced voltage (Fig. 6i).

Due to the high specific surface area and powerful electrostatic sensing capabilities, ISW could also identify larger and faster moving objects in a non-contact manner (Fig. 7a). ISW can be well camouflaged in tree branches to monitor UAV flight and human movement in field environments (Fig. 7b, c). Compared with flying perpendicular to the ISW, the UAV induced a higher voltage in the ISW when flying in a direction parallel to the ISW, due to the larger induction area of the parallel flight. When UAV was hovering, some continuous and irregular

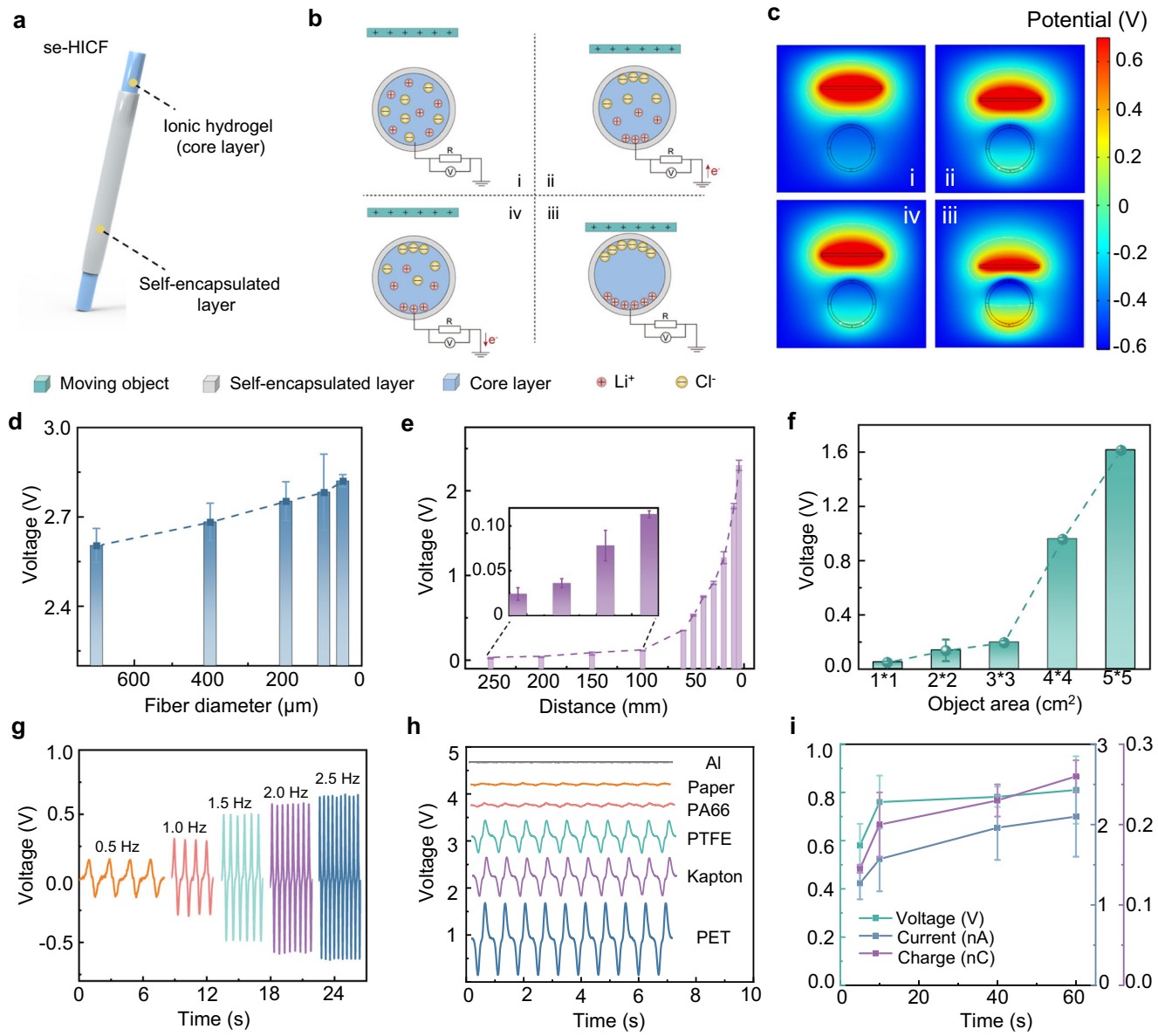

**Fig. 5 | Noncontact sensing performance of se-HICFs. a** Schematic illustration about se-HICFs as self-powered sensor based on the principles of electrostatic induction (**b**) The working principle of se-HICFs for noncontact sensing. **c** The potential distribution during the se-HICFs sensing moving object simulated by the COMSOL software based on finite-element simulation. **d** The $V_{oc}$ of se-HICFs with different diameters in sensing the test object. **e** The $V_{oc}$ of se-HICFs in different sensing distance with the test object. **f** The $V_{oc}$ of se-HICFs when sensing test objects with different sizes. **g** The $V_{oc}$ of se-HICFs when sensing the PET film with area of $5 \times 5$ cm$^2$ with frequency from 0.5 to 2.5 Hz. **h** The $V_{oc}$ of se-HICFs when sensing different test objects with various materials. **i** The $V_{oc}$ of se-HICFs in diameter of 50 μm and 200 μm when sensing a PET film within 60 days. (The se-HICFs length is 10 cm, the PET film is in size of $5 \times 5$ cm$^2$, the loading distance is 8 cm and the reference distance is 1 cm). The error bar for each data point in (**d–i**) is standard deviation calculated based on 3 parallel measurements.

small peaks would be generated due to the rotations of the propeller blade in the air. The ISW also monitored passing people with different motion patterns, and the generated signals clearly identified whether the passers-by were running, walking or wandering (Fig. 7d, e). In addition, ISW can maintain strong sensing capabilities even on rainy days. In the simulated rainy-day test using a sprinkler device, water droplets had no obvious impact on the monitoring process of the ISW, and the movements of butterfly and human could still be recognized, demonstrating its strong ability to adapt to climate change (Supplementary Fig. 17). ISW's powerful sensing capability for moving objects and its imperceptibly camouflage performance in the natural environment make it has broad application prospects in fields of smart agriculture, environmental protection and even homeland security reconnaissance.

## Discussion

In the present work, we proposed a stress-induced adaptive phase transition strategy for the facile fabrication of self-encapsulated ionic hydrogel fibers. Taking us-IHs with long/short polymer complementarity and dynamic physical cross-linked networks as precursor material, se-HICFs can be prepared on a large scale by direct stretching or dip-drawing molding. The unique sheath-core structure of se-HICFs formed by phase transition endowed the fibers with high stability, excellent mechanical strength and superb electrostatic induction ability similar to single-electrode triboelectric nanogenerators. Inspired by nature, se-HICFs could be further woven into a bionic spiderweb structure and camouflaged in the natural environment, realizing the non-contact 3D depth perception of moving objects such as insects, UAV and humans. Looking forward, combined with machine

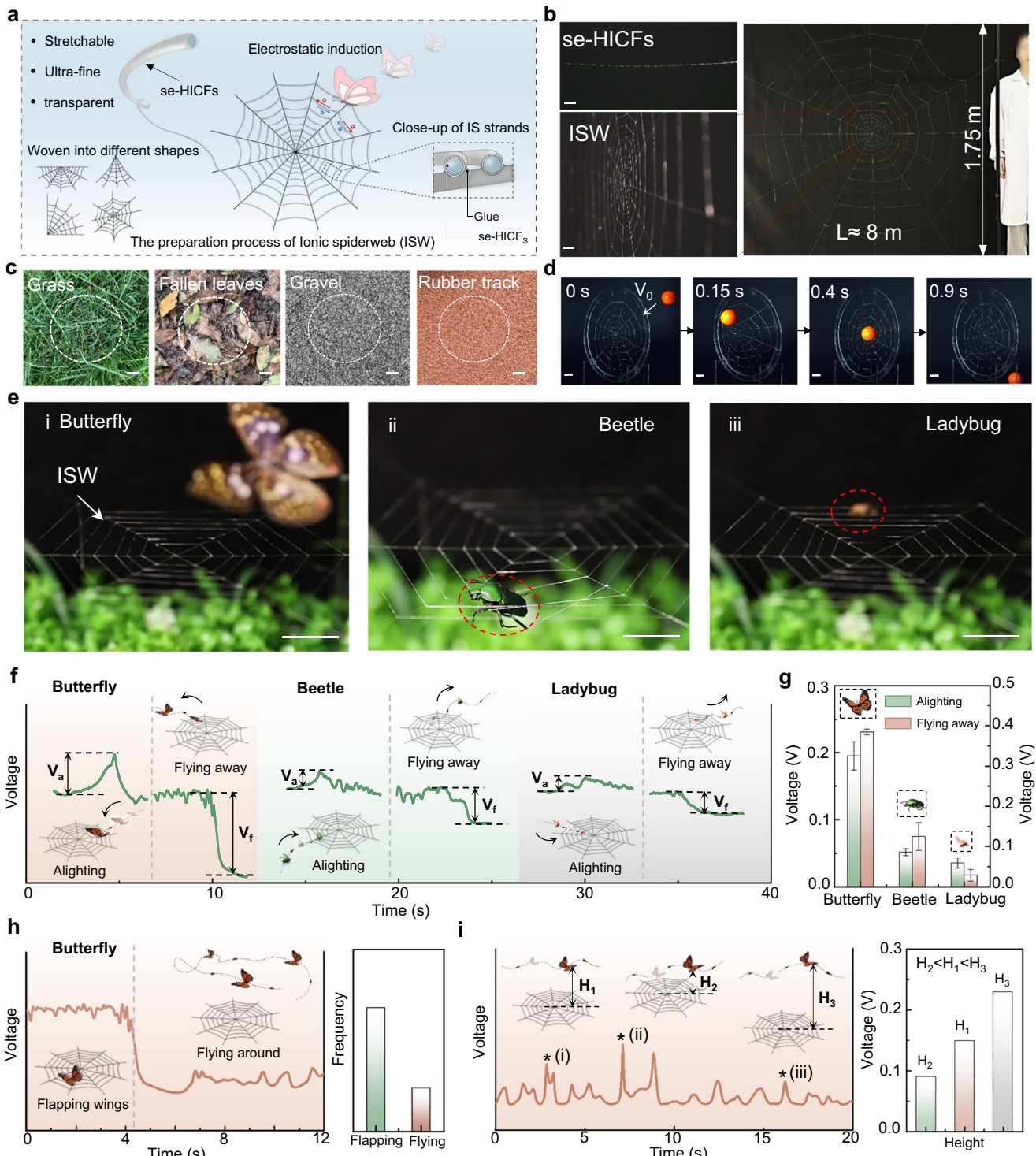

**Fig. 6 | The fabrication of Ionic spider web (ISW). a** Schematic diagram about preparation process of the ISW. **b** A huge ISW with a diameter of 1.7 m fabricated by se-HICF with a length of ~8 m. Scale bar: 2 cm. **c** ISW could camouflage on grass, fallen leaves, gravel and rubber track. Scale bar: 2 cm. **d** The high-speed digital video camera recorded that the ISW could resist the ball with high velocity. Scale bar: 1 cm. **e** Photographs of butterfly, beetle and ladybug alighting on the ISW. Scale bar: 2 cm. **f, g** The electrical signal and induced voltage of alighting, flying away the ISW for butterfly, beetle and ladybug. **h** The induced voltage of flapping wings on the ISW and flying around the ISW for butterfly. **i** The induced voltage of different flight height for butterfly. The error bar for each data point in (**g**) is standard deviation calculated based on 3 parallel measurements.

learning, widely deployed and distributed biomimetic sensors based on se-HICFs can achieve high-precision localization, recognition, and motion feature monitoring for targets, which can be applied to pest monitoring in smart agriculture and camouflage reconnaissance for defending homeland security. On the other hand, further research is urgently needed to construct an integrated sensing system including sensors, signal processing circuits, communication modules to realize wireless passive sensing in the wild. It is believed that the stress-induced adaptive phase transition strategy and the prepared se-HICFs will inspire the design of next-generation flexible electronics.

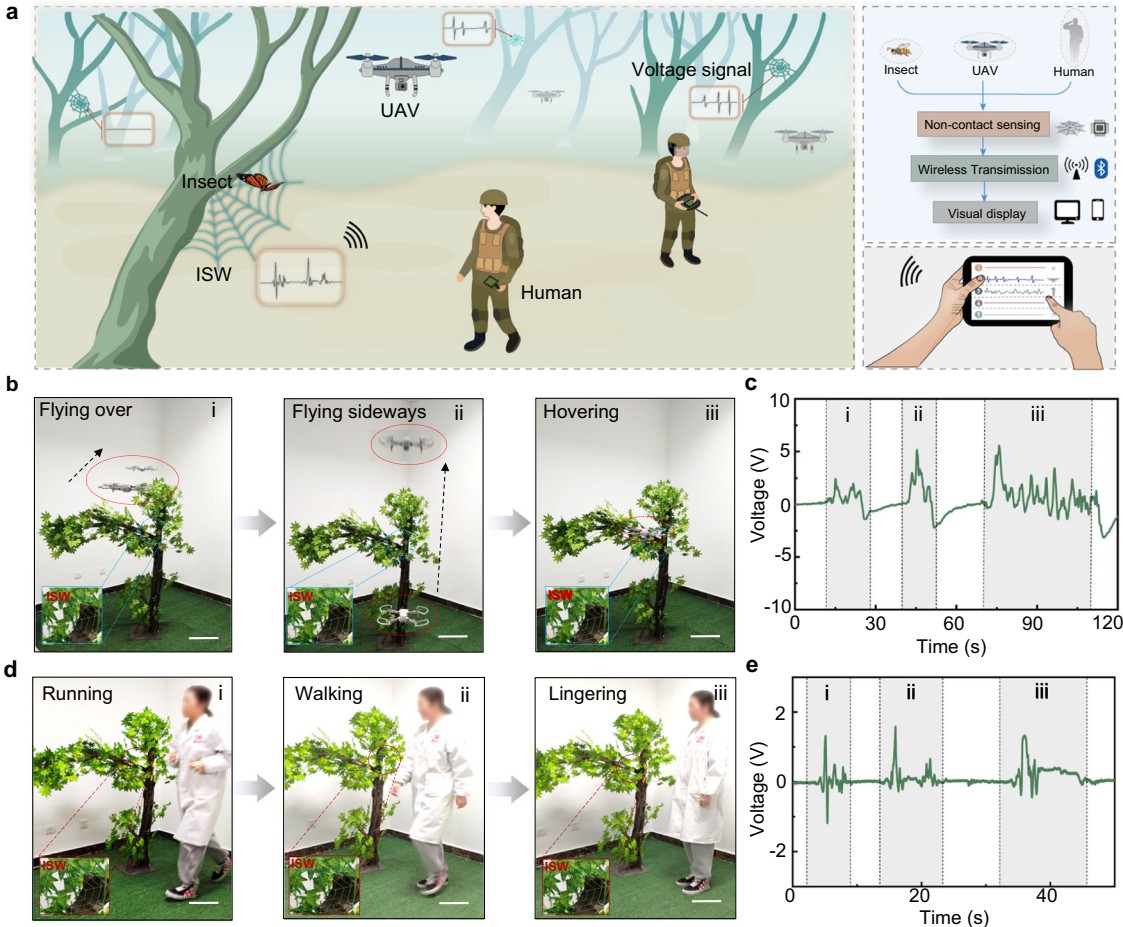

**Fig. 7 | se-HICFs for camouflage monitoring system. a** Schematic diagram about the ISW's applications in camouflage monitoring. **b** Photographs of UAV passing through the ISW in different flight modes. Scale bar: 30 cm. **c** The electrical signals induced by UAV passing through the ISW. **d** Photographs of human passing through the ISW with different movements. Scale bar: 30 cm. **e** The electrical signals induced by human passing through the ISW.

## Methods

### Materials
Poly (vinyl alcohol) (PVA, CAS: 9002-89-5, 1799, P105126) and branched polyethylenimine (PEI, CAS: 9002-98-6, MW = 600, E808878) were purchased from Aladdin. Lithium chloride (LiCl, CAS: 7447-41-8, L812571) was purchased from Macklin.

### Synthesis of the us-IHs
The us-IHs was fabricated through a one-pot and freezing/thawing method. Briefly, 2.0 g of PVA, 1.0 g of PEI, and 4.0 g of LiCl were dissolved in 9 mL of deionized water to form a homogeneous solution in a 95 °C water bath with vigorous stirring for 1–2 h. Then, the uniform sol was molded into film by glass mold and then frozen at −20 °C overnight and transferred to room temperature for 6 h.

### Synthesis of the se-HICFs
The as-fabricated us-IHs was cut into a dumbbell shape and stretched by ESM301/Mark-10 system with a tensile speed of 30 mm·min⁻¹ into hydrogel fibers. Alternatively, us-IHs was first heated to a molten state, and then needles were immersed in it for dip-drawing molding. Through the self-encapsulating process, the se-HICFs with sheath core structure were formed.

### Characterization of us-IHs and Se-HICFs
Fourier transform infrared spectroscopy (FTIR) was obtained using a VERTEX80v spectrometer (Bruker, Karlsruhe, Germany). X-ray photoelectron spectra (XPS) were recorded by PHI 5000 VersaProbe III with

aluminum Kα source (1848.6 eV) and a collimator at 15 kV and 50 W. X-ray diffraction (XRD) was performed using PANalytical X'Pert3 Power with 2θ ranged from 5° to 80°, where Cu Kα radiation ($\lambda = 1.5406$ Å) were operated at 40 kV and 40 mA. Thermal Gravimetric Analyzer (TGA, SDT-Q600, TA Instruments) was analyzed in air atmosphere with a heating rate of 10 K·min⁻¹ and a temperature of 25–500 °C.

### Measurement of us-IHs
Tensile test of the ionic hydrogel was performed by ESM301/Mark-10 system at room temperature. Rheological property test of the us-IHs was performed by Huck rotating rheometer (MCR92, Anton Paar, Austria).

### Measurement of se-HICFs
Tensile test of the se-HICFs were performed by ESM301/Mark-10 system with stretching rate 100 mm·min⁻¹ at room temperature. The optical images of the se-HICFs compared with hair were taken by a metallomicroscope (Nikon LV100ND). Scanning electron microscopy (SEM) and Energy Disperse Spectroscopy (EDS) images of the se-HICFs were obtained using a Hitachi field emission scanning electron microscope (SU 8020, Hitachi, Tokyo, Japan). The photographs of ISW holding the moving ball were obtained by the High-Speed Digital Video Camera (Mini AX, FASTCAM, Japan).

### Electrical measurement
The resistance of the ionic hydrogels with different LiCl content were measured by CHI 660E electrochemical workstation. EIS spectra were

obtained at a range of 1 MHz to 0.1 Hz with an AC amplitude of 10 mV. A linear motor (LinMot E1100) was used to control the movement of statically induced materials to test the non-contact electrostatic induction capability of se-HICFs. The voltage, current, and charge transfer were measured by an electrometer (Keithley 6517B) and recorded by an oscilloscope (Teledyne LeCroy HD 4096).

The author affirm that human research participants provided informed consent for publication of the images in Fig. 6b, 7d.

## Reporting summary
Further information on research design is available in the Nature Portfolio Reporting Summary linked to this article.

## Data availability
All data supporting the findings described in this manuscript are available within the paper and its Supplementary Information, and the source data is available on the public data repository Figshare (https://doi.org/10.6084/m9.figshare.24893280). Source data are provided with this paper.

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

## Acknowledgements

This work was financially supported by the National Natural Science Foundation of China (T2125003 and 81971770 to Z. Li; 52372174 to D. Luo; 82102231 and 82372141 to Z. Liu.), Natural Science Foundation of Beijing Municipality (JQ20038 and L212010 to Z. Li), Fundamental Research Funds for the Central Universities (EOEG6802X2 and E2E46806 to Z. Li), National Key Research and Development Program of China (2022YFB3804703 and 2022YFE0111700 to Z. Li; 2021YFB3201204 and 2022YFB3205602 to D. Luo), the Fundamental Research Funds for the General Universities to Z. Li, and the Fundamental Research Funds for the General Universities to Z. Liu.

## Author contributions

Y.L., C.W. and Z.Liu. contributed equally to this work. Z.Li., D.L., and Y.L. were responsible for the experimental concept and design. Y.L., C.W., and Z.Liu. designed the study, performed experimental measurement and data analyses. Y.L. and D.L. wrote the manuscript. X.C.Q. and J. H. took and processed the experimental photograph and video. J.T.X. contributed to COMSOL simulation. S.Y.C. contributed to the explanation of working mechanism of device. Y.S.G. contributed to schematic of the article. Y.X.W. and Y.S.L. contributed to the investigation of literatures. Z.Li. was responsible for project administration, conceptualization, supervision, funding acquisition, validation and review. All authors discussed the results and commented on the manuscript.

## Competing interests

The authors declare no competing interests.
