## [Peer Review File · Nature Communications]

REVIEWER COMMENTS

Reviewer #1 (Remarks to the Author):

Reviewer :

In this work, the authors have developed a stress-induced adaptive phase transition strategy for fabrication of self-encapsulated ionic hydrogel fibers. The sheath-core structure of se-HICFs endowed the fibers with high stability, excellent mechanical strength and superb electrostatic induction ability similar to single-electrode triboelectric nanogenerators. Based on the non-contact 3D depth perception of moving objects, it can achieve high-precision localization, recognition, motion feature monitoring for targets. This work shows high novelty, and the material design is ingenious and impressive. The following comments should be well addressed before consideration of publication.

1. About Figure 1a, the authors stated that "the stress-induced phase transition and evaporation of water molecules on the surface of the ionic fiber facilitated the transformation of surface layer to form a tight encapsulation layer ". Please explain if this water-soluble property of hydrogel fiber affects its application as biomimetic sensors.
2. The stability, the mechanical strength, superb electrostatic induction under different humidity conditions should be provided. It seems the fibers should be affected by humidity, but the authors do not state the humidity conditions for the characterization. The authors should add this part in this paper. How about the effect of the monitoring the movements of insects, UVA, and humans in natural environments when in a rain day?
3. Error bars should be provided in Figures 3g, h, i, and Figure 4h.
4. In Figure 5f, j, k, l, the scale bars should be presented more clearly.
5. The supplementary figure in the manuscript should be named Fig. S8, Fig. S9. Please check the manuscript and Supplementary Material and modify it.
6. The authors claimed stress-induced stretching/dip-drawing of us-His into ionic fibers, but how to address the limitations of preparation in large-scale production and commercialization.
7. Some relevant literatures about stress-induced stretching/dip-drawing forming fibers and biomimetic sensors are suggested to be cited and discussed in the introduction part. (e.g., Nat. Comm., (2019) 10:5293, Adv Mater. 2022; e2201843, Adv. Fiber. Mater. 3, 107–116 (2021), Adv. Fiber. Mater. 4, 319–320 (2022))

Reviewer #2 (Remarks to the Author):

In this work, the authors proposed a stress-induced adaptive phase transition strategy to fabricate functional hydrogel-based ionically conductive fibers. These fibers can be obtained by direct stretching or dip-drawing from ionic hydrogel with ultra-stretchable networks without any organic solvent and equipment. Simultaneously, the self-encapsulation triggered by adaptive phase transition enhanced mechanical strength significantly and endowed the fibers with long-term reliability. Finally, these fibers could be woven into delicate spider web structures and camouflaged in the wild environments for high spatiotemporal resolution 3D depth-of-field sensing of different moving medias based on non-contact electrostatic induction. I am delighted to review this paper in great depth as it represents a very interesting study. Based on the experimental findings, the proposed scheme shows promising in addressing the challenges of developing a convenient large-scale fabrication strategy and improving the stability of functional fibers during service. This strategy to prepare fibers will contribute to the design of next-generation flexible electronics. Considering its novelty and potential impact, I believe this work represents a noteworthy advancement in the field

of flexible sensors. Therefore, I recommend its publication in Nature Communications with minor revisions.

1. The force-strain curves in Fig. 3b showed the force applied to the fibers was closed to zero during the third and fourth stretching stages. Why did it happen? The authors should explain it.
2. According to the test of multi-stretching, the diameters of fibers were 846, 380, 88 and 20 μm under different stretching stages, which were corresponded to the stretching ratio were 71, 2556, 107352 and 1932336 in theory, respectively. Compared with the changes in stretching ratio, the changes in fiber diameter are not very significant, the authors should explain it.
3. The authors mentioned that when the $\tan \delta$ of hydrogel was in the range of 0.6 to 1.1, suggested it was the highly spinnable. Why is this range suitable for spinning?
4. For the part about description of the mechanism of non-contact electrostatic induction, the authors should give more detailed explanation to help readers understand. In the current version, we are quite confused why PET is positively charged at start.
5. Why this sensing is called 3D depth-of-field sensing? What makes it unique compared to other sensing works? The authors should give more explanations.
6. For the part of ISW sensing the moved human and UVA, what's the authors' basis for distinguishing the signals generated by different movements?
7. The authors mentioned "molten hydrogel" in the introduction. Does this mean heating and melting the sample? Can it be replaced by gel state us-IH, is it more suitable?
8. We noticed that the scale numbers on some graphs are shifted, as shown in Figure 3e. This will cause misunderstanding for readers, please correct it.
9. The figures in the manuscript are not clear enough. The author should upload higher-resolution images.

Reviewer #3 (Remarks to the Author):

This manuscript presents the development of self-encapsulated hydrogel-based ionically conductive fibers (se-HICFs). The authors demonstrate the potential applications of these fiber materials in single electrode mode triboelectric nanogenerators for camouflage sensing in the form of a spider web. However, it is important to note that similar hydrogel fibers with comparable raw materials, structures, design principles and properties have been previously reported by the authors in *Adv. Mater.* (2022, 34, 2105416). Furthermore, the method of fiber fabrication has also been described in an earlier publication in *J. Mater. Chem. A* (2021, 9, 10240). Additionally, when compared to other hydrogel fibers investigated in recent studies (*ACS Nano* 2020, 14, 11, 14929–14938; *Nat Commun* 14, 1370 (2023); *Nat Commun* 13, 3369 (2022)), the mechanical, electrical, and sensing performances of our se-HICFs do not exhibit significant improvements. The demonstrated applications in this work are not unusual and can be achieved using different materials or device systems. Furthermore, if the connection to the spider web for camouflage sensing is not delicate,

the connection and back-end system would not be effectively camouflaged. Consequently, the novelty and significance of the present work are limited, making it unsuitable for publication in this journal.

Response to Reviewer 1

General Comment: In this work, the authors have developed a stress-induced adaptive phase transition strategy for fabrication of self-encapsulated ionic hydrogel fibers. The sheath-core structure of se-HICFs endowed the fibers with high stability, excellent mechanical strength and superb electrostatic induction ability similar to single-electrode triboelectric nanogenerators. Based on the non-contact 3D depth perception of moving objects, it can achieve high-precision localization, recognition, motion feature monitoring for targets. This work shows high novelty, and the material design is ingenious and impressive. The following comments should be well addressed before consideration of publication.

Response: We sincerely appreciate the Reviewer for the encouraging comments, recommendations, and detailed suggestions regarding the improvement of the manuscript. We have carefully addressed the reviewer's question as follows.

Comment 1. About Figure 1a, the authors stated that "the stress-induced phase transition and evaporation of water molecules on the surface of the ionic fiber facilitated the transformation of surface layer to form a tight encapsulation layer ". Please explain if this water-soluble property of hydrogel fiber affects its application as biomimetic sensors.

Response: We highly appreciate the reviewer's comment. In our study, stress-induced shrinkage of the polymer network of hydrogel fibers results in directional migration of water molecules to the outside of the fiber and subsequent evaporation on the fiber surface. Once the stress-induced phase transition occurs, the structure of the self-encapsulated layer becomes relatively stable and less susceptible to damage even in high-humidity environments. This is due to the fact that dehydration will cause the polymer to transform into a glassy state, resulting in strongly restricted segmental motions; in addition, during the phase transition process, the intermolecular hydrogen bonds between polymer molecules and water molecules will also be converted into intramolecular hydrogen bonds of polymer molecules. Therefore, even if it is a hydrophilic polymer, the water solubility will be greatly reduced after forming a sheath structure. As you suggested, in order to confirm that the presence of water-soluble polymers in self-encapsulating ionic fibers does not affect their application as biomimetic sensors, we have

immersed the fibers in water to verify their stability. Notably, the self-encapsulated fibers did not undergo swelling or structural collapse even after 60 min of immersion, and still maintained great mechanical properties, demonstrating the robustness and stability of the sheath-core structure. We have added relevant experiments to the revised manuscript.

The revised contents are as follows:

[Main Text: Page 11, Line 280 to Page 12, Line 282] [Supplementary Information: Page 6]

Even so, se-HICFs did not undergo swelling or structural collapse after being immersed in water for 60 minutes, and maintained great mechanical properties, highlighting the robustness and stability of the self-encapsulated layer (Fig. S11).

Figure S11. (a) The photography of fiber immersing in the water after 60 min. (b) The fiber during the stretching test.

Comment 2. The stability, the mechanical strength, superb electrostatic induction under different humidity conditions should be provided. It seems the fibers should be affected by humidity, but the authors do not state the humidity conditions for the characterization. The authors should add this part in this paper. How about the effect of the monitoring the movements of insects, UVA, and humans in natural environments when in a rain day?

Response: We greatly appreciate the reviewer's comment. We have added the experiments on the stability, mechanical properties and electrostatic induction capabilities of self-encapsulated fibers under different humidity conditions (RH: 45~85 %). Additionally, we have also demonstrated the monitoring of moving insects and humans in a natural environment on rainy day (it is not possible to

monitor the movement of UVA in simulated rainy weather, as the UVA is not waterproof and could not be flown in rainy conditions). These two parts have been added in the revised manuscript.

The revised contents are as follows:

[Main Text: Page 11, Line 275-280] [Supplementary Information: Page 6]

Additionally, we also assessed the stability of the se-HICFs in different humidity environments⁴⁶. The increase in relative humidity resulted in enhanced ductility, reduced tensile strength, and higher elongation of se-HICFs (Fig. S10); However, even at high humidity (RH 85%), se-HICFs still had great mechanical property with the tensile strength of 16.8 MPa and a large tensile deformation of 843%.

Fig. S10. The stress (a) and strain (b) of the fibers under different humidity on the test of stress-strain curve.

[Main Text: Page 13, Line 314-316] [Supplementary Information: Page 7]

Furthermore, se-HICFs also exhibited great electrostatic induction properties even in high-humidity environments (RH 85%), revealing their stability and broad application prospects as non-contact self-powered sensors (Fig. S14).

Fig. S14. The stability of electrostatic induction ability for se-HICFs under different humidity conditions.

[Main Text: Page 14, Line 353-357] [Supplementary Information: Page 8]

In addition, ISW can maintain strong sensing capabilities even on rainy days. In the simulated rainy-day test using a sprinkler device, water droplets had no obvious impact on the monitoring process of the ISW, and the movements of butterfly and human could still be recognized, demonstrating its strong ability to adapt to climate change (Fig. S16).

Fig. S16. (a) Simulation of a rainy day. The V_{oc} of ISW sensing the moving (b) butterfly and (c) human.

Comment 3. Error bars should be provided in Figures 3g, h, i, and Figure 4h.

Response: We highly appreciate the reviewer's professional comment. We have added error bars in Figure 3g, h, i and Figure 4h.

The revised contents are as follows:

[Main Text: Page 24, Figure 3g, h, i]

Figure 3 Stress-induced stretching/dip-drawing of us-I into ionic fibers. (g) Effects of the needle's diameters, (h) hydrogel solution temperatures and (i) drawing speed. (j) Photograph of ionic fibers after draw spinning.

[Main Text: Page 26, Figure 4h]

Figure 4 The stress-induced adaptive phase transition process and mechanical properties of se-HICFs. (h) The variation of se-HICF's water content within 40 days.

Comment 4. In Figure 5f, j, k, l, the scale bars should be presented more clearly.

Response: We highly appreciate the reviewer's insightful and professional comment. We have redrawn the scale bars in Figure 3f, j, k and l to make them more clearly.

The revised contents are as follows:

[Main Text: Page 24, Figure 5f, j, k, l]

Figure 3 Stress-induced stretching/dip-drawing of us-IHs into ionic fibers. (f) Photographs during dip-drawing spinning of the ionic fibers with different needles. **Scale bar: 2 cm.** (j) Photograph of ionic fibers after draw spinning. **Scale bar: 2 cm.** (k) Photograph shows the great stretchability of ionic fibers. Scale bar: 1 cm. (l) Photographs of beads-on-a-string structure in ionic fibers. Scale bar: 2 mm.

Comment 5. The supplementary figure in the manuscript should be named Fig. S8, Fig. S9. Please check the manuscript and Supplementary Material and modify it.

Response: We highly appreciate the reviewer's comment. We have carefully checked and modified them in the manuscript and Supplementary Material.

Comment 6. The authors claimed stress-induced stretching/dip-drawing of us-His into ionic fibers, but how to address the limitations of preparation in large-scale production and commercialization.

Response: We highly appreciate the reviewer's comment. For the time being, large-scale preparation of hydrogel-based fibers has focused on spinning technologies, including wet/dry/gel spinning, electrostatic spinning. However, the above processes still face some challenges: for example, the poor spinnability of hydrogels and their precursor solutions makes it difficult to mold meter-sized or longer

fibers; in addition, the high solvent consumption and cumbersome production processes of these processes are not in line with the trend towards green chemistry.

Considering the above issues, in this work we developed a stress-induced adaptive phase transition strategy to prepare self-encapsulating ionic fibers. Since the ultra-stretchable ionic hydrogels, as precursors, have an ultimate tensile deformation of more than 1,900,000 times, ultrafine fibers with a length of tens of meters can be obtained by direct stretching centimeter-scale ionic hydrogel at room temperature, without using additional organic solvents. Meanwhile, taking into account the future needs of large-scale production and commercial preparation, we further developed the dip-drawing process for molten gels and systematically explored the effects of different process parameters on fiber morphology. In addition, the dip-drawing molding has also been demonstrated to be able to use multiple needles to synchronously fabricate meter-scale, high-quality self-encapsulating fibers and wind them on reels for easy storage, which greatly improves the efficiency of fiber production. It is worth mentioning that the dip-drawing process also does not require complicated spinning equipment, and self-encapsulating fibers can be prepared using modified existing dip coating devices.

In summary, this manuscript provides a potential alternative to overcome the problems of poor spinnability of raw filaments, cumbersome process, high energy consumption, and environmental unfriendliness in the traditional fiber manufacturing process. In our future work, we will optimize devices and manufacturing processes for the economical, efficient and industrial production of ionic fibers.

Comment 7. Some relevant literatures about stress-induced stretching/dip-drawing forming fibers and biomimetic sensors are suggested to be cited and discussed in the introduction part. (e.g., Nat. Comm., (2019) 10:5293, Adv Mater. 2022; e2201843, Adv. Fiber. Mater. 3, 107–116 (2021), Adv. Fiber. Mater. 4, 319–320 (2022))

Response: We appreciate the reviewer’s comment. We have read these papers carefully, which greatly helped improve our manuscript. All related studies have been cited in the revised manuscript.

The revised contents are as follows:

[Main Text: Page 17, Line 427-428; Page 19, Line 481-484, Line 495-496]

5. Pan, S. and Zhu M. Nanoprocessed silk makes skin feel cool. *Advanced Fiber Materials* **4**, 319-320 (2022).
31. He, W. *et al.* A protein-like nanogel for spinning hierarchically structured artificial spider silk. *Advanced Materials* **34**, 2201843 (2022).
32. Dou, Y. *et al.* Artificial spider silk from ion-doped and twisted core-sheath hydrogel fibers. *Nature Communications* **10**, 5293 (2019)
38. Lv, S. *et al.* Flexible humidity sensitive fiber with swellable metal–organic frameworks. *Advanced Fiber Materials* **3**, 107-116 (2021).

Response to Reviewer 2

General Comment: In this work, the authors proposed a stress-induced adaptive phase transition strategy to fabricate functional hydrogel-based ionically conductive fibers. These fibers can be obtained by direct stretching or dip-drawing from ionic hydrogel with ultra-stretchable networks without any organic solvent and equipment. Simultaneously, the self-encapsulation triggered by adaptive phase transition enhanced mechanical strength significantly and endowed the fibers with long-term reliability. Finally, these fibers could be woven into delicate spider web structures and camouflaged in the wild environments for high spatiotemporal resolution 3D depth-of-field sensing of different moving medias based on non-contact electrostatic induction. I am delighted to review this paper in great depth as it represents a very interesting study. Based on the experimental findings, the proposed scheme shows promising in addressing the challenges of developing a convenient large-scale fabrication strategy and improving the stability of functional fibers during service. This strategy to prepare fibers will contribute to the design of next-generation flexible electronics. Considering its novelty and potential impact, I believe this work represents a noteworthy advancement in the field of flexible sensors. Therefore, I recommend its publication in Nature Communications with minor revisions.

Response: We sincerely appreciate the Reviewer for the encouraging comments, recommendations, and detailed suggestions regarding the improvement of the manuscript. We have carefully addressed the reviewer's question as follows.

Comment 1. The force-strain curves in Fig. 3b showed the force applied to the fibers was closed to zero during the third and fourth stretching stages. Why did it happen? The authors should explain it.

Response: We thank the reviewer for his/her comment. During the third and fourth stretching stages, the centimeter-scale ionic hydrogel had transformed into fibers with diameters of 88 μm and 20 μm , respectively, which were able to withstand relatively small forces (about 0.02 N, in the present work). Thus, when putted in comparison with the larger forces applied to macroscopic hydrogel samples during stretching, the forces applied to the fibers appeared to be close to zero, but was not zero. Similar phenomena have also appeared in other reports (e.g. *Adv. Mater.* 2019, 31, e1904029). When multiple

stages of stretching were used to overcome the limitations of instrument travel distance, force-stretching ratio curves showed that the force experienced by the polymer network were relatively small in the final stretching stage.

Comment 2. According to the test of multi-stretching, the diameters of fibers were 846, 380, 88 and 20 μm under different stretching stages, which were corresponded to the stretching ratio were 71, 2556, 107352 and 1932336 in theory, respectively. Compared with the changes in stretching ratio, the changes in fiber diameter are not very significant, the authors should explain it.

Response: We highly appreciate the reviewer for his/her professional and significant comment. In general, only for ideally elastic materials, the cross-sectional area changes in a fixed proportion with the tensile ratio. However, hydrogels, consisting of polymers and a large amount of water, exhibit viscoelastic behavior. When subjected to external forces, energy dissipation occurs in the polymer network through changes in chain conformation, untangling of physical entanglements, sliding of segments, spatial displacement of polymers chains, reversible cross-linking and noncovalent domain unfolding (*Chem. Rev.* 2021, 121, 4309-4372). Due to the complexity and variability of structural transformations within the polymer network during stretching, there may not be a fixed proportional relationship between the change in hydrogel cross-sectional area and the stretching ratio. Therefore, in the present work, although the tensile ratio varied considerably, the change in fiber diameter was not significant.

Comment 3. The authors mentioned that when the $\tan \delta$ of hydrogel was in the range of 0.6 to 1.1, suggested it was the highly spinnable. Why is this range suitable for spinning?

Response: We highly appreciate the reviewer's comment. The $\tan \delta$ value is fundamentally related to the chain entanglement effect, which is a critical parameter to distinguish whether it is suitable for fluid spinning. A recent study showed that solutions or sols with $\tan \delta$ value ranging from 0.6 to 1.5 have great spinning capabilities due to the appropriate viscoelastic behavior (*Nano Lett.* 2022, 22, 9396-9404). However, $\tan \delta$ value higher than 1.5 and lower than 0.6 show elastic dominant behavior

and viscous dominant behavior respectively, both of which are not conducive to spinning.

Comment 4. For the part about description of the mechanism of non-contact electrostatic induction, the authors should give more detailed explanation to help readers understand. In the current version, we are quite confused why PET is positively charged at start.

Response: We apologize for the unclear description. The reason why PET is positively charged is that it exhibits weaker electronegativity than se-HICF. We had tested the triboelectric series of PET and se-HICFs according to the developed method (*Matter* 2023, 6, 1–17). When compared to highly electronegative material (PTFE), PET was less electronegative than se-HICF (Figure R1). Therefore, during the contact/separation process between se-HICF and PET, PET will be positively charged. In this paragraph, PET was only used as an example of a positively charged object to facilitate a more intuitive description of the mechanism, and is not intended to emphasize the chemical composition of the object. In order to avoid readers' misunderstanding, we have changed it to “a moving object with positive charges” in the revised manuscript. Moreover, we have also provide an expanded explanation of the mechanism of non-contact electrostatic induction to facilitate the readers' understanding.

Figure R1. Triboelectric output performance when se-HICF and PET were in contact with/separated from PTFE respectively. The results show that PET exhibited weaker electronegativity than se-HICF.

The revised contents are as follows:

[Main Text: Page 12, Line 290-298]

When a moving object with positive charges approached the se-HICF, the negative induced charges appeared on the near-object side of self-encapsulated layer, Li^+ ions were absorbed to the side of core layer close to the negative charges, and Cl^- ions moved to the other side; in this process, free electrons flowed into the se-HICF to shield the net charge, which generated an output current signal as it passed through the external circuit. Conversely, when a positively charged object moved away from the se-HICF, the ion displacement tended to be reversed, and free electrons flowed out of the se-HICF to generate a opposite current in the circuit. Repeating this process produced a pulsed electrical signal.

Comment 5. Why this sensing is called 3D depth-of-field sensing? What makes it unique compared to other sensing works? The authors should give more explanations.

Response: We greatly appreciate the reviewers' comments. 3D depth-of-field sensing refers to the technology and capability of a sensing system to accurately measure and perceive the depth or distance information of objects within a scene. In other words, depth-of-field sensing can not only collect the information projected by the object to be measured on a two-dimensional plane, but also sense the spatial orientation of the object in the three-dimensional scene, especially in the depth direction.

In our work, the non-contact depth sensing performance of ionic conductive spiderweb is reflected in that it can not only distinguish the motion information of target objects of different sizes on the same two-dimensional plane, but also determine whether the target object is approaching or leaving the sensor in three-dimensional space. In addition, se-HICFs-based non-contact sensing have some unique advantages: (i) High customizable and environmentally adaptable: sensors based on flexible stretchable fibers could be programmatically woven into 2D or 3D biomimetic structures with different size and shapes to meet monitoring requirements. (ii) Strong camouflage: se-HICF has an ultra-fine, transparent and stretchable structure similar to spider silk. Spiderweb structures prepared with se-HICF can be delicately camouflaged in the natural environment. (iii) Low operating energy consumption: se-HICF realizes depth-of-field sensing based on the principle of non-contact electrostatic induction, which does not consume energy during the signal generation stage. (iv) Lightweight and low cost with potential for scale manufacturing.

Comment 6. For the part of ISW sensing the moved human and UVA, what's the authors' basis for distinguishing the signals generated by different movements?

Response: We highly appreciate the reviewer's comment. From the signal waveform point of view, the most significant difference between the signals generated by human and UVA motion is that there is a set of small peaks with similar frequencies in the induced voltage signal generated by UVA due to the regular rotation of the propeller. This is the characteristic signal that distinguishes the UVA. On the other hand, the motion of the two objects can also be analyzed from the perspective of signal threshold. Since biomimetic spiderweb sensors are usually hung on branches, only objects flying at high altitudes are more likely to approach the spiderweb, thereby obtaining stronger induced voltage signals. Looking to the future, by combining big data and machine learning, the widely deployed distributed ionic conductive spider web can enable high-precision localization, identification, and motion feature monitoring for moving targets.

Comment 7. The authors mentioned “molten hydrogel” in the introduction. Does this mean heating and melting the sample? Can it be replaced by gel state us-IH, is it more suitable.

Response: We appreciate the reviewer's comment. The molten hydrogel refers to the sol state hydrogel formed after heating. Considering the process requirements of dip-drawing molding, it is difficult to use gel state us-IHs to fabricate ionic fibers.

Comment 8. We noticed that the scale numbers on some graphs are shifted, as shown in Figure 3e. This will cause misunderstanding for readers, please correct it.

Response: We are very grateful to the reviewer for his/her careful reading and pointing out our shortcomings. As you suggested, we have adjusted the position of the scale numbers on graphs to make them easier for readers to understand.

Comment 9. The figures in the manuscript are not clear enough. The author should upload higher-resolution images.

Response: Thank you for your kind reminder. In the revised manuscript, all Figures have been replaced with high-resolution versions.

Response to Reviewer 3

This manuscript presents the development of self-encapsulated hydrogel-based ionically conductive fibers (se-HICFs). The authors demonstrate the potential applications of these fiber materials in single electrode mode triboelectric nanogenerators for camouflage sensing in the form of a spider web. However, it is important to note that similar hydrogel fibers with comparable raw materials, structures, design principles and properties have been previously reported by the authors in *Adv. Mater.* (2022, 34, 2105416). Furthermore, the method of fiber fabrication has also been described in an earlier publication in *J. Mater. Chem. A* (2021, 9, 10240). Additionally, when compared to other hydrogel fibers investigated in recent studies (*ACS Nano* 2020, 14, 11, 14929–14938; *Nat Commun* 14, 1370 (2023); *Nat Commun* 13, 3369 (2022)), the mechanical, electrical, and sensing performances of our se-HICFs do not exhibit significant improvements. The demonstrated applications in this work are not unusual and can be achieved using different materials or device systems. Furthermore, if the connection to the spider web for camouflage sensing is not delicate, the connection and back-end system would not be effectively camouflaged. Consequently, the novelty and significance of the present work are limited, making it unsuitable for publication in this journal.

Response: We appreciate the reviewer for his/her comments and efforts towards improving our manuscript. The reviewer referred to five papers from three perspectives to compare our work with existing studies. However, this comparison just proves that our work embodies the highlights from three perspectives of the five studies mentioned above, emphasizing the advantages and importance of our work in different aspects. It is worth noting that our work is not a simple accumulation of multiple highlights; rather, it stems from innovations in preparation theories and strategies, thereby achieving the fabrication of delicate self-encapsulated structures, and ultimately realizing the application in scenarios that best suit its characteristics.

In addition, we have three **highlights** in this study, which can be listed as follows:

(i) Theory and strategy of stress-induced adaptive phase transition

The proposal of this strategy solves the problems faced by current spinning technology for preparing hydrogel-based fibers, such as poor spinnability of hydrogels and their precursor solutions, high solvent consumption, and cumbersome production processes. Meanwhile, taking into account the future needs of large-scale production and commercial preparation, we further developed the dip-drawing process for molten gels, which was proven to be able to use multiple needles to synchronously

fabricate meter-scale, high-quality fibers and wind them on reels for easy storage. Compared with traditional methods, our strategy significantly improves the efficiency of fiber production and reduces costs.

(ii) Facile fabrication of self-encapsulating structures

Stress-induced adaptive phase transition could directly obtain self-encapsulated structures, greatly improving the stability of the hydrogel-based fiber. Compared with the traditional wrapping, covering and coating strategy, this method not only simplifies the processing steps, but also solves the phenomena of peeling and abnormal fracture caused by the modulus mismatch and weak interfacial adhesion between the core and encapsulation layers.

(iii) Biomimetic camouflage

The sheath-core structure formed by self-encapsulation that endows se-HICFs with strong electrostatic induction capability. It could be further woven into a biomimetic spiderweb structure for efficient non-contact 3D depth-of-field camouflage sensing.

Taken together, our in-depth exploration and innovation in mechanisms, methods, and applications provide an example for the preparation of high-performance and low-cost self-encapsulating ionic hydrogel fibers, which is expected to inspire the design of next-generation flexible electronics.

As always, we highly respect the reviewers' suggestions. The reviewers listed five very important references, and we have cited the above reports in the revised manuscript. In addition, through careful investigation of the above-mentioned studies, we found that these works are fundamentally different from our report in terms of their scientific connotations, which, to some extent, rather highlights the novelty and significance of our work. In order to eliminate the reviewer's misunderstanding, we will describe the highlights and features of this work as follows from the materials, preparation methods, performance, and other aspects mentioned by the reviewer.

Adv. Mater. 2022, 34, 2105416

As a preface, we would first like to clarify that we are not reporting on a study of hydrogel synthesis, but rather developing a novel theory and methodology for the preparation of self-encapsulating ionic fibers based on stress-induced adaptive phase transitions. Compared with the previously reported work (*Adv. Mater.* 2022, 34, 2105416), the present study only used similar raw material compositions but was completely different in terms of fabrication methods, characteristics and physicochemical

properties, micro- and nanoscale structures, working mechanisms, and application scenarios.

Specifically, in a previous report, a versatile ionic hydrogel with fast self-healing ability and stable conductivity in cryogenic environment was synthesized, which could mimic the structure and function of myelinated axons, exhibit fast and potential-gated signal transmission characteristics, and ultimately achieve high-fidelity and high-throughput information interaction for robots. However, in this study, we reported a stress-induced adaptive phase transition strategy to prepare functional ionic fibers simply, efficiently, and environmentally friendly through simple stretching/dip-drawing molding. In addition, we also systematically discussed the effects of different chemical synthesis parameters and stress-induced phase transition process conditions on the morphology, micro-nano structure, chemical structure, mechanical properties, electrical properties and non-contact sensing capabilities of the prepared ionic hydrogel fibers. Inspired by nature, the acquired self-encapsulating fibers (se-HICFs) can be further woven into biomimetic spiderweb structures to realize high-temporal-spatial resolution 3D depth-of-field camouflage sensing for different moving objects in wild environments. To demonstrate the differences more intuitively, the highlights of previous work and this study are summarized in **Table R1**.

Table R1. Systematic comparison between this study and Adv. Mater. 2022

	Adv. Mater. 2022, 34, 2105416	This Study
Fabrication Strategy	Simple mold forming	Stress-induced adaptive phase transition (Stretching/Dip-drawing molding)
Main Characteristics	 1. Self-healing 2. Anti-freezing 	 1. Self-encapsulating (sheath-core structure) 2. Spinnability
Structural Analogs	Artificial nerve fiber 	 1. Morphology: similar to spider silk 2. Function: similar to stretchable single-electrode triboelectric nanogenerator

		Device Features	 1. Stretchable and self-healing in extremely cold environment 2. High fidelity and throughput transmission 	 1. Powerful electrostatic induction capability 2. High customizability and environmental adaptability 3. Strong camouflage ability 4. Self-powered, high stability (derived from sheath-core structure)
Working Mechanism	Signal transmission in a capacitance model	Non-contact electrostatic induction
Application Scenarios	Communication unit for information transmission and energy delivery in robot	Biomimetic sensor for efficient non-contact 3D depth-of-field camouflage sensing

J. Mater. Chem. A 2021, 9, 10240

In this work (*J. Mater. Chem. A* 2021, 9, 10240), the hydrogel fibers were prepared by a draw-spinning method using polyacrylic acid hydrogel cross-linked with silica nanoparticles. The as-prepared fiber exhibited a breaking strength of 79 MPa and a breaking strain of 65.8%. The authors improved water resistance and mechanical properties of fibers through the coating of carbon nanotube sheath, and explored the effects of different wrapping angles between carbon nanotubes and hydrogel fibers on fiber shrinkage or elongation.

In comparison, our work tells a completely different story. It is our optimization of the chemical structure of ionic hydrogels with super-stretchable networks (us-IHs) as precursor materials that ionic fibers could be prepared by direct stretching or dip-drawing from molten hydrogels at room temperature without the requirement for high energy consumption, cumbersome production processes and large amounts of solvent consumption. More importantly, our proposed stress-induced adaptive

phase transition strategy could directly prepare self-encapsulating fibers with sheath-core structures in one step. In the work mentioned by the reviewer (*J. Mater. Chem. A* 2021, 9, 10240), the authors first need to use chemical vapor deposition to prepare a carbon nanotube forest, then extract the aligned carbon nanotube sheets, and finally coat them on the surface of the hydrogel fiber, which required at least three steps. In our report, stress promoted the directional migration and evaporation of water molecules in the optimized hydrogel, and enabled the ionic fibers to achieve self-encapsulation through the phase transition of the surface layer, which formed a sheath-core structure with more than 1,200 times enhanced mechanical strength, endowing the fibers with long-term reliability. To demonstrate the differences more intuitively, the highlights of previously reported work and this study are summarized in **Table R2**.

Table R2. Systematic comparison between this study and *J. Mater. Chem. A* 2021

	J. Mater. Chem. A 2021, 9, 10240	This Study
Fabrication Strategy	Draw-spinning	Stress-induced adaptive phase transition (Stretching/Dip-drawing molding)
Preparation of fiber encapsulation layer	Multi-step process 1. Grow carbon nanotube forest by chemical vapor deposition 2. Extract the aligned carbon nanotube sheets 3. Coat on the surface of hydrogel fibers	One-step process Self-encapsulation (stress-induced adaptive phase transition)
Fiber-based device types	Actuator based on twisted fibers	Biomimetic sensor based on self-encapsulated fibers
Working Mechanism	Water-moisture driven contraction	Non-contact electrostatic induction
Application Scenarios	Moisture meters and moisture sensitive smart windows	Non-contact 3D depth-of-field camouflage sensing

ACS Nano 2020, 14, 14929; Nat. Commun. 2023, 14, 1370; Nat. Commun. 2022, 13, 3369

The work published in ACS Nano (*ACS Nano* 2020, 14, 14929-14938) described a method for the continuous engineering production of electro-responsive hydrogel fibers via a self-lubricated spinning strategy, achieving a tensile fracture stress of 5.6 MPa and a strain of 159%. As a typical electroactive polymer, the hydrogel fiber possessed a fast and stable electroactuation performance under electric field. The electrochemical-responsive ionic hydrogel fiber was capable of acting as soft robots underwater to mimic biological motions.

The work published in Nature Communications (*Nat. Commun.* 2023, 14, 1370) introduced an aqueous pultrusion spinning process to continuously produce hydrogel microfibers with a high Yong's modulus of 428 MPa, an elongation of 219%, and a toughness of 19.8 MJ/m³ under ambient conditions. In addition, due to moisture sensitivity, the hydrogel fibers possessed an ultra-rapid self-healing, high damping, and supercontraction properties, which have a great potential in miniature soft machines and robots.

In another work published in Nature Communications (*Nat. Commun.* 2022, 13, 3369), the authors introduced a macromolecule conformational shaping strategy that enabled mechanical programming of hydrogel fiber. The elongation ratio of the fiber ranged from 105% ± 2% to 2630% ± 120%, and the corresponding tensile strength ranged from 1210 ± 120 kPa to 47 ± 5 MPa. The authors demonstrated that hydrogel microfibers with different macromolecule conformations can be built-in layered formats, enabling various hydrogel electronic applications, such as large strain and ultrafast responsive fiber sensors in robotic bird, large deformations and anti-freezing helical electronic conductor, and large strain capable Janus springs energy harvesters in wearables.

Through comparison, it is not difficult to find that our reported work differs significantly from the above three references in terms of fiber preparation method, structure, properties and application areas. As mentioned above, our work emphasizes the mechanism and strategy of preparing highly stable self-encapsulating ionic fibers through stress-induced adaptive phase transition, and ultimately realizes the application in non-contact depth-of-field camouflage sensing. Therefore, around this goal, we consider more whether the fiber preparation method is simple, efficient, and environmentally friendly, and whether the fiber structure and performance meet the requirements of spinnability, camouflage ability, high stability, and high electrostatic induction capability. In our humble opinion, it is unfair to some extent to simply compare the performance of materials with different components, structures and

application fields in a certain aspect. The key issue is to assess whether these parameters can serve the needs of the application scenario well. By way of analogy, just as we cannot require biomimetic sensors to have high tensile strength like advanced fabrics and high conductivity like conductors. In other words, if the above assumptions hold true, then I believe that such fibers will not reach the excellent level of the biomimetic sensor reported in this article in terms of spinnability, camouflage ability and depth-of-field sensing performance.

To sum up, the three works mentioned by the reviewers and our study propose different strategies to fabricate hydrogel-based fibers with different structures, characteristics and functions that have been applied in soft robots or sensors. Each study has its own advantages and highlights. The comparison of these four works is shown in **Table R3**.

Table R3. Systematic comparison between this study and ACS Nano 2020, Nat. Commun. 2022, Nat. Commun. 2022

	ACS Nano 2020, 14, 14929-14938	Nat. Commun. 2023, 14, 1370	Nat. Commun. 2022, 13, 3369	This Study
Fabrication Strategy	Two-step UV initiated self-lubricated spinning	Pultrusion spinning	Wet spinning	Stress-induced adaptive phase transition (Stretching/Dip-drawing molding)
Encapsulation of fibers	No encapsulation layer	No encapsulation layer	No encapsulation layer	Self-encapsulation (stress-induced adaptive phase transition)
Mechanical property	5.6±0.6 MPa, 159%	428 MPa, 219%	240±30 kPa~ 2050±370 MPa, 105±2%~630±120%	46 MPa, 423%
Electrical property	Not mentioned	Not mentioned	88.7 S/m	3.6 S/m
Working mechanism	Electroactuation under electric field	Moisture sensitivity	Macromolecule conformational shaping strategy	Non-contact electrostatic induction
Application Scenarios	Soft robots: electrically triggered to accomplish various predefined manners	Soft machines and robots (prospect): high strength, crack resistance, self-healing	Fiber sensors Large strain and fast response Helical conductors large deformations, and anti-freezing Janus springs energy harvesters large strain	Biomimetic sensor high-temporal-spatial resolution 3D depth-of-field camouflage sensing for different moving media in the field environment.

In summary, we have developed a novel stress-induced adaptive phase transition strategy to achieve efficient, simple, and environmentally friendly preparation of self-encapsulating ionic fibers. The prepared fibers have strong weaving ability and can be made into bionic spider web-like sensors to be hidden in the natural environment. In response to the reviewer's concerns about connection and back-end system camouflage performance, there are a series of mature solutions: (1) this biomimetic sensor can be connected to commercially available miniature signal acquisition and wireless transmission module (typically only about 2 square centimeters in size). This miniaturized, highly integrated unit can match the natural environment in color and shape, making it more difficult to be detected. (2) Circuits connected to biomimetic sensors can be sheltered, such as hidden underground, without affecting its sensing function. Finally, our work further confirms that se-HICFs-based biomimetic sensors can perform high spatiotemporal resolution 3D depth-of-field sensing of different moving media such as insects, UVA, and humans, which holds significant promise in smart agriculture and homeland security. We greatly appreciate the valuable opportunity provided by the reviewer to illustrate the importance of this study and sincerely hope that our reply will clear up the reviewer's misunderstanding.

REVIEWERS' COMMENTS

Reviewer #1 (Remarks to the Author):

The authors revised the paper according to my comments, and now it can be published.

Reviewer #2 (Remarks to the Author):

I would like to support the publication of this revised manuscript

Response to Reviewer 1

Comment: The authors revised the paper according to my comments, and now it can be published.

Response: We sincerely appreciate the Reviewer for the encouraging comments, recommendations, and detailed suggestions regarding the improvement of the manuscript.

Response to Reviewer 2

Comment: I would like to support the publication of this revised manuscript.

Response: We sincerely appreciate the Reviewer for the encouraging comments, recommendations, and detailed suggestions regarding the improvement of the manuscript.